# Spinal mechanisms and feasibility of Dry Needling versus Botulinum Toxin Type A in post-stroke lower limb spasticity: A proof-of-concept randomized clinical trial protocol (STROKE-POC)

Clara Pujol-Fuentes[1,2], Bart Eeckhaut[3], Samuel Fernández Carnero[4], Daniel Fernández Sanchís[2,5], Theodore Wein[6], Pablo Herrero[2,7*], Wim Saeys[3], Mindy F. Levin[8,9]

1 Faculty of Health and Sciences, Department of Physiotherapy, Universidad Europea de Valencia, Valencia, Spain, 2 Instituto de Investigación Sanitaria de Aragón, Zaragoza, Spain, 3 Faculty of Health Sciences and Medicine, Department of Rehabilitation Sciences and Physiotherapy, University of Antwerp, Antwerp, Belgium, 4 Universidad de Alcalá, Facultad de Medicina y Ciencias de la Salud, Departamento de Enfermería y Fisioterapia, Grupo de Investigación Fisioterapia y Dolor, Alcalá de Henares, Spain, 5 Faculty of Health Sciences, Universidad San Jorge, Villanueva de Gállego, Zaragoza, Spain, 6 Faculty of Medicine, Department of Neurology and Neurosurgery, McGill University, Montreal, Quebec, Canada, 7 Faculty of Health Sciences, Department of Physiotherapy, Universidad de Zaragoza, Zaragoza, Spain, 8 McGill University, School of Physical and Occupational Therapy, Montreal, Quebec, Canada, 9 Center for Interdisciplinary Research in Rehabilitation of Greater Montreal (CRIR), Montreal, Quebec, Canada

* pherrero@unizar.es

## Abstract

### Introduction

Stroke often causes spasticity, impacting mobility and quality of life. Botulinum Toxin type A (BTX-A) and Dry Needling (DN) are treatments that reduce spasticity, although Botulinum Toxin type A injections can cause adverse effects. No studies have directly compared their effects at spinal, muscular, functional, quality-of-life, and cost-effectiveness levels. This study aims to determine the spinal mechanisms of BTX-A and DN on post-stroke lower limb spasticity, while also assessing feasibility, safety, and exploratory effects at muscular, functional, quality-of-life, and cost-effectiveness levels.

### Methods and analysis

This is a protocol of a proof-of-concept, feasibility randomized clinical trial including 90 participants from Canada, Belgium, and Spain who experienced a first stroke in the previous 12 months and present plantar flexor spasticity. Time since stroke (0–12 months) will be recorded and explored as a potential modifier of treatment response. Participants will be randomly assigned to receive either one session of BTX-A or 12 weekly sessions of DN. Blinded evaluators will assess outcomes before, during, and after treatment, with a 4-week follow-up. The primary outcome will be spinal

**Data availability statement:** The study protocol and supporting documentation are publicly available on Zenodo at https://doi.org/10.5281/zenodo.20034064 (CC BY 4.0). Participant-level data are not yet available as data collection has not commenced. Anonymized data will be deposited in the same repository upon study completion.

**Funding:** Our study spans three countries - Canada, Belgium, and Spain - and involves a team of experts from relevant fields aligned with the study's scope. After a rigorous selection process that included input from multiple researchers, our study was awarded funding through the NEURON ERA-NET 2022 call following an independent peer review. This support is backed by the European Union's Horizon 2020 program (GA 964215) and key partner organizations, namely Instituto de Salud Carlos III (ISCIII) for the project "AC22/00016" (Spain), FRQS Fonds de recherche du Québec - Santé (FRQS) #324226 (Canada), and Fonds Wetenschappelijk Onderzoek (FWO) #49317 (Belgium). Additional co-funding was provided by the European Union-Next Generation EU/ PRTR.

**Competing interests:** The authors have declared that no competing interests exist.

mechanisms of spasticity, measured using the Tonic Stretch Reflex Threshold and its velocity sensitivity. Secondary outcomes will assess: a) muscular architecture and echotexture (measured with ultrasound); b) muscle tone/resistance using the Modified Ashworth Scale; c) gait and mobility (instrumented analysis, Timed Up and Go, 10-Meter Walk Test); d) muscle strength with dynamometry; e) quality of life with the EuroQoL questionnaire; and f) cost-effectiveness (analytic model). The findings will provide preliminary data to inform a future definitive trial.

## Ethics and Dissemination

This research project has secured funding from the NEURON ERA-NET 2022 call, supported by the European Union's Horizon 2020 program (GA 964215) and co-funded by the European Union-Next Generation, and has undergone peer review. Ethical approval has been obtained from Spain, Canada, and Belgium. The study is registered in ClinicalTrials.gov (NCT06296082) and the Clinical Trials Information System (CTIS) under the number 2024-510866-18-00. The study protocol is registered on Zenodo (https://doi.org/10.5281/zenodo.20034064)

## Clinical Trials

Clinical Trials NCT06296082; https://clinicaltrials.gov/study/NCT06296082

## Introduction

Stroke is the third-leading cause of death and disability worldwide [1], imposing a substantial social and economic burden on patients, families, and healthcare systems [2]. After a stroke, a significant percentage of patients develop spasticity [3]. Ankle spasticity increases fall risk [4] and reduces QoL [5].

Botulinum Toxin type A (BTX-A) infiltration is currently considered the gold standard for managing post-stroke spasticity [6], while the clinical use of Dry Needling (DN) remains limited despite emerging evidence supporting its potential benefits [7]. BTX-A inhibits acetylcholine release at the neuromuscular junction, producing a temporary chemical denervation that reduces muscle overactivity and spasticity [8]. Consistent with this mechanism, BTX-A has demonstrated clinical effectiveness in reducing post-stroke spasticity and improving functional outcomes and gait [6]. DN has also demonstrated effectiveness in reducing spasticity in neurological populations, including post-stroke patients, as reported in a recent meta-analysis [7]. DN is proposed to act at the neuromuscular endplate, similar to BTX-A, but via mechanical rather than chemical mechanisms [9]. Experimental histological studies in animal models have shown that although DN induces minor mechanical injury to muscle fibers after repeated needle insertions, the tissue follows the normal sequence of muscle regeneration, which was nearly complete within approximately one week [9]. Beyond these local effects, preliminary neurophysiological studies suggest that DN may also influence spinal excitability [10]. However, no randomized trial has directly

compared their effects on impairment (muscular and spinal mechanisms), functional activity levels (gait and mobility), quality of life, and cost-effectiveness.

BTX-A injection has been associated with adverse effects on muscle, such as lingering atrophy, changes in elasticity [11], incomplete histological recovery [12], and reduction in muscle thickness and pennation angle of the gastrocnemius [13]. In contrast, no adverse effects have been reported for DN, except for those common to needling techniques, such as bleeding or hematoma. Regarding costs, one session of BTX-A treatment in Spain costs from 603.64€ to 707.59€ per patient annually [14] while a session of DN treatment costs 14,96€ [15]. However, since costs differ in different countries and the number of sessions for each treatment is different, it is necessary to determine their comparative cost-effectiveness.

This study aims to investigate the spinal mechanisms of BTX-A and DN in reducing post-stroke spasticity in plantar flexor muscles. Specific objectives are 1) to explore their impact on spasticity at the muscle, spinal cord, and functional activity levels as well as on quality of life; 2) to assess their cost-effectiveness, and 3) to evaluate their safety and clinical feasibility.

## Materials and methods

### Design

This is a prospective multicenter, proof-of-concept feasibility randomized controlled trial in Spain, Canada, and Belgium. It will include two parallel groups, a multiple-baseline design, and an assessor-blinded approach. The trial follows the SPIRIT 2025 guideline and the WHO Trial Registration Data Set (S1 File) [16], as well as Good Clinical Practice guidelines and the Declaration of Helsinki.

### Patient population

Inclusion Criteria: 1) aged 18–75 years; 2) post-stroke spasticity in ankle plantar flexors (Modified Ashworth Scale (MAS) scores of 1–2); 3) first stroke; 4) 0–12 months evolution; 5) no previous BTX-A or DN treatment for spasticity; 6) ankle passive range of motion ≥20° (approximately) with knee flexion ~30°; 7) independent ambulation with or without aids. Exclusion Criteria: 1) medical conditions interfering with data interpretation; 2) contraindications for BTX-A or DN treatment; 3) changes in anti-spasticity medication dosage (if appropriate), either during the trial or within the 3 months before participation; 4) pregnant or breastfeeding.

### Recruitment strategy

Spain will recruit the participants at Hospital Universitario Lozano Blesa, Canada at Hospital networks associated with McGill University and the University of Montreal, and Belgium at the REVARTE rehabilitation center and AZ Voorkempen Hospital. All patients will provide written informed consent. Appointment reminders will be sent by phone or email, and efforts will be made to collect outcome data even if participants discontinue the assigned intervention.

### Interventions to be measured

Before participant enrollment, a rehabilitation physician or neurologist will review clinical records to verify eligibility and rule out contraindications. Subsequently, a physiotherapist will confirm the spasticity grade (MAS 1–2) and ankle passive range of motion ≥20° with knee flexion ~30°. Participants will be informed that no additional BTX-A or DN treatments will be permitted during the study.

Eligible participants will complete three baseline assessments [17], participate in weekly assessment sessions, and undergo a follow-up assessment four weeks later (Fig 1).

| | Study Period | | | | | | | | | | |
|---|---|---|---|---|---|---|---|---|---|---|---|
| | Enrollment | Allocation | Postallocation | | | | | | | | Closeout |
| Time point | -t1 | 0 | t1 | t2 | T3 | t4 to t8 | t9 | T10 to t14 | t15 | T16 to t18 | t19 |
| **Enrollment:** | | | | | | | | | | | |
| Eligibility screen | X | | | | | | | | | | |
| Informed consent | X | | | | | | | | | | |
| Allocation | | X | | | | | | | | | |
| **Interventions:** | | | | | | | | | | | |
| Intervention BTX-A | | | | | X | | | | | | |
| Intervention DN | | | | | X | X | X | X | X | | |
| **Assessments:** | | | | | | | | | | | |
| TSRT | | | X | X | X | X | X | X | X | | X |
| US | | | X | X | X | X | X | X | X | | X |
| Muscle resistance | | | X | X | X | X | X | X | X | | X |
| Gait analysis | | | X | | | | X | | X | | X |
| TUG + 10MWT | | | X | | | | X | | X | | X |
| Muscle strength | | | X | X | X | X | X | X | X | | X |
| QoL | | | X | | | | | | X | | X |
| Cost form | | | | | | | | | | | X |
| Patient acceptance | | | | | | | | | | | X |
| Post-treatment assessor's blinding questions | | | | | | | | | | | X |
| Sample size, recruitment, and consent rates for future RCTs. | | | | | | | | | | | X |

**Fig 1. SPIRIT diagram for the schedule of enrollment, interventions and evaluation.** This figure shows the timeline of participant enrollment, allocation, interventions, and outcome assessments according to the SPIRIT guidelines.

Regarding the treatment, BTX-A will be injected once by a clinician, whereas DN will be performed once per week for 12 weeks by an experienced physiotherapist in dry needling. This treatment schedule is consistent with previous clinical studies on dry needling for post-stroke spasticity, which have used repeated sessions over several weeks to achieve sustained effects [18]. Moreover, a once-weekly interval allows adequate time for tissue recovery between sessions, in accordance with proposed muscle repair timelines following needling interventions [9]. Mandatory muscles for treatments are the medial gastrocnemius, the lateral gastrocnemius, the soleus, and the tibialis posterior. Optional muscles are the flexor digitorum longus and brevis, the flexor and extensor hallucis longus [19].

The BTX-A group will receive onabotulinumtoxinA (Botox®, Allergan) with mandatory muscles getting 300 units and optional muscles getting up to 100 additional units (maximum dose of 400 units) delivered with a 27-gauge (0.45 mm) beveled needle [19]. Localization of the involved muscles for BTX-A will be guided by either ultrasound (US) and/or electromyography (EMG), according to the clinician´s preference, to ensure correct intramuscular placement and avoid neurovascular structures.

For the DN group, solid, filiform, non-beveled stainless-steel needles (Agupunt®, 0.30×40 to 0.30 x 75 mm) will be used, depending on the necessary needle length to reach the target muscle. The physiotherapist specialized in DN will diagnose trigger points in neurological patients, as described in the literature for DNHS (Dry Needling for Hypertonia and Spasticity) technique [20,21]. The needle will be advanced until a local twitch response (LTR) is observed as a

confirmatory sign in the case of DN, repeating short insertions until the LTR is exhausted or the participant's tolerance limit is reached. In the case of BTX-A injection, it will be injected near the endplate area, without the need of provoking LTRs.

Both interventions will be guided in real time by B-mode ultrasound using a portable Butterfly iQ+ device (Butterfly Network, Inc.) to ensure accurate identification of target muscles, needle trajectory monitoring, and safety control. When clinically required, electromyography (EMG) guidance may also be used to verify muscle activation and optimize needle placement accuracy in either BTX-A or DN sessions. Local anesthesia will not be used for any procedure. After each intervention session, participants will remain under observation for at least 15 minutes to detect immediate adverse events.

For both BTX-A and DN procedures, patient positioning will follow standardized criteria based on the treated muscle: the medial and lateral gastrocnemius and the flexor hallucis longus will be treated in prone position; the soleus, tibialis posterior, flexor digitorum longus, and brevis in lateral decubitus on the affected side (or supine if clinically indicated); and the extensor hallucis longus in supine position (tibialis posterior can also be treated in supine, according to the clinician´s preference). If participants are unable to tolerate these positions, alternative postures will be allowed based on comfort and documented in the case report form [20,21]

A standardized Standard Operating Procedure (SOP) was jointly developed by the three participating countries to harmonize intervention procedures and ensure consistency across sites. This document consolidates and standardizes the processes already described in this protocol, particularly those related to muscle identification and the specific techniques for injection and needling. The SOP includes visual diagrams and procedural checklists to enhance reproducibility and safety verification. Before recruitment began, all clinicians participated in joint training sessions and practical demonstrations to standardize muscle localization, needle handling, and aseptic technique. The SOP also defines a safety verification process at each visit, including pre-treatment eligibility checks and post-session monitoring for adverse reactions. Procedural consistency among clinicians is periodically assessed through cross-site reviews and inter-rater evaluations, complementing the independent oversight provided by the Data Monitoring Committee. These measures collectively reinforce methodological transparency, reproducibility, and procedural uniformity across all study centers.

Treatment will be discontinued or modified in case of any serious adverse event related to the intervention, a clinically significant worsening of the participant's condition as judged by the site investigator, or at the participant's request. Any discontinuation or modification will be documented and reported to the Data Monitoring Committee and the Ethics Committees.

To promote adherence, participants will receive written schedules and appointment reminders by phone or email before each visit. Attendance at DN sessions and any deviations from the assigned schedule will be recorded in dedicated case report forms to monitor compliance with the intervention protocol. Participants will continue to receive usual rehabilitation care such as physiotherapy and occupational therapy as prescribed by their clinical teams. No additional antispasticity pharmacological treatments (e.g., oral baclofen or other botulinum toxin injections) will be allowed during the trial period.

**Sample size calculation**

90 participants (30 per country) will be recruited following the Recommendations for Planning Pilot Studies in Clinical and Translational Research [22,23]. Each group in each country will have 15 patients. This proof-of-concept feasibility randomized controlled trial is not primarily designed to test definitive efficacy hypotheses, but rather to obtain reliable estimates of recruitment, retention, adherence, safety, and preliminary effect sizes to inform the design and sample size of a future definitive trial.

In line with these recommendations, the sample size was chosen to ensure reasonable precision in estimating variability in the primary outcome (Tonic Stretch Reflex Threshold, TSRT) and other key feasibility parameters, rather than to achieve a specific level of statistical power for hypothesis testing. As a plausibility check, assuming a moderate between-group effect size for change in TSRT (Cohen's $d \approx 0.5$), a two-sided comparison with an alpha level of 0.05 and 80% power would typically require on the order of 60–70 participants in total. The planned sample of 90 participants, therefore,

provides a conservative margin that allows for potential attrition and the multicenter structure of the trial, while still fulfilling its primary feasibility objectives.

## Clinical significance

Although this is a feasibility study, the data obtained on mechanisms, safety, patient acceptance, and cost-effectiveness will provide valuable preliminary information to inform larger randomized controlled trials and contribute to the development of patient-centered approaches in spasticity management. In terms of translation to clinical practice, a patient-centered approach is crucial, as some patients may prefer a non-pharmacological approach for post-stroke spasticity [24], which has lower costs and fewer adverse effects. Additionally, understanding the mechanisms of action of each approach in stroke could benefit other neurological conditions such as Parkinson's Disease, Multiple Sclerosis, Spinal Cord Injury, or Traumatic Brain Injury.

## Outcomes

**Primary outcomes.** Spasticity will be measured with the Tonic Stretch Reflex Threshold (TSRT) and its velocity sensitivity ($\mu$) [25–27]. This quantitative assessment will be performed using the Montreal Spasticity Reflex Threshold (MSRT) system, developed at the Centre for Interdisciplinary Research in Rehabilitation (CRIR, Montreal, Canada). This device records the joint angle at which muscle activity begins in response to passive stretch, providing a precise and objective measure of spasticity. The ankle plantarflexors will be tested in supine position with the knee flexed approximately 30°, using an electrogoniometer aligned 2 cm below the lateral malleolus and surface EMG electrodes placed over the medial gastrocnemius (agonist) and tibialis anterior (antagonist). The physiotherapist will apply passive ankle dorsiflexion movements at different velocities, as instructed by the MSRT software, to estimate the TSRT and its velocity sensitivity. Safety precautions will include ensuring no electrical interference, removing wristwatches and phones from the testing area, and securing cables to prevent motion artifacts.

The MSRT system has been previously used to measure TSRTs in stroke populations. Studies have demonstrated its reliability and clinical applicability in determining the degree of post-stroke spasticity, as well as its construct validity as a motor-control–based measure that also distinguishes spasticity from other forms of hypertonia [27–29]. These findings support the use of TSRT-derived measures as objective and physiologically grounded outcomes in clinical research.

**Secondary outcomes.** They will be measured: 1) Muscle architecture [30] and echotexture [31] with ultrasound imaging, using a portable B-mode system (Butterfly iQ +, Butterfly Network, Inc.) with a 50% gain and 5 cm depth. Participants will be seated with the knee flexed at 90°, and the probe placed at 30% of the distance between the medial tibial condyle and the medial malleolus at a 35–45° angle relative to the horizontal plane. Three transverse and three longitudinal images will be acquired for each participant by a physiotherapist trained in musculoskeletal ultrasound; 2) resistance to passive stretching with MAS [32]. Participants will lie in a supine position, and the examiner will manually move the ankle into dorsiflexion at a consistent speed, maintaining hand contact at the calcaneus and along the plantar surface of the foot to control the movement and perceive the resistance level; 3) gait analysis will be conducted with two precise 3D motion capture systems [33] depending on the country: Xsens system (MVN Awinda, Movella, Hendersen, USA) [34] will be used in Canada and Belgium, and the Move Human Sensors MoCap System [35] will be used in Spain. Participants will walk at a self-selected speed over a 10-meter walkway, with inertial sensors placed on the pelvis, thighs, shanks, and feet to capture kinematic data and spatiotemporal gait parameters; 4) functional mobility and walking speed will be assessed by the 10 Meter Walk Test (10MWT) [36] and the Timed Up & Go (TUG) test [37], both administered by a physiotherapist; 5) muscle strength will be measured with a hand-held dynamometry (MicroFET 2) [38]. Plantarflexion will be tested in seated position with 90° of knee flexion to minimize coactivation, with the dynamometer placed against the metatarsal heads and stabilized against the floor; dorsiflexion will be tested with the examiner holding the device manually in the same position; 6) QoL will be evaluated using the EuroQoL-5D questionnaire [39] and 7) the economic impact of the

interventions will be analyzed through a cost-effectiveness assessment by computing the Incremental Cost-Effectiveness Ratio (ICER) expressed in €/QALY. Direct and indirect costs related to treatment, medication, and patient care will be collected throughout the study, including self-reported expenses such as transportation, medical visits, complementary tests, and healthcare resource use. The analysis will compare the clinical effectiveness of BTX-A and DN relative to their associated costs. A non-parametric bootstrapping approach will be used to estimate and represent the uncertainty surrounding the ICER distribution. The final cost-effectiveness evaluation will be performed at the end of the study, although costs will be continuously collected during the intervention period.

Additionally, we will estimate the sample size, recruitment, and consent rates for future RCTs, while safety and feasibility of each intervention will be continuously monitored throughout the trial.

### Masking

Assessments will be conducted by blinded assessors for treatment allocation. Participants and staff cannot be blinded due to the nature of the intervention but are instructed not to disclose their treatment allocation. Data will be anonymized, entered by an independent researcher into secure databases, and analyzed by a blinded statistician. Code breaks will only be allowed in crucial cases for patient management. Assessors will later report if they correctly guessed the patient group allocation, evaluating the effectiveness of blinding.

To minimize potential performance and detection bias, outcome assessments are performed by blinded evaluators, and data analysis, including ultrasound image processing and instrumented gait analysis, is conducted by researchers blinded to treatment allocation. Importantly, most primary and secondary outcomes are objective and instrument-based (e.g., TSRT, ultrasound imaging, instrumented gait analysis, dynamometry), reducing susceptibility to expectation-related bias once standardized acquisition protocols are followed. The Modified Ashworth Scale, although examiner-dependent, is administered under standardized conditions and constitutes a secondary outcome

### Randomization

Patients will be stratified by spasticity severity, sex, and country and block-randomized (1:1) into 15 pairs per site and assigned to one of the two interventions. The allocation sequence will be generated using a computer-based randomization module within the REDCap platform (Research Electronic Data Capture, Vanderbilt University Medical Center, Nashville, TN, USA).

Allocation concealment will be ensured through centralized electronic assignment within REDCap, such that the allocation sequence is not accessible to recruiting clinicians prior to participant enrollment.

### Patient and public involvement

The project aims to improve evidence-based practice through a Public and Patient Involvement (PPI) approach, which will take into account the patients' preferences and expectations (patient-centered approach).

To ensure successful patient engagement and feedback, and to disseminate the results, a Scientific Expert Committee, a Patient Advisory Group, a dedicated website space, and a platform for exchanging experiences will be established. The patients' acceptance will be assessed through a questionnaire.

Each country will have its own group, and it will be done separately for BTX A and DN. The consortium meetings will include consultation with the Patient Advisory Group to get feedback throughout the study.

### Statistical analyses

Descriptive statistics will summarize baseline demographic and clinical characteristics by treatment group. Continuous variables will be presented as means and standard deviations or medians and interquartile ranges, as appropriate, while

categorical variables will be described using frequencies and percentages. Normality will be assessed using the Shapiro–Wilk test and visual inspection of histograms.

The primary analysis will focus on changes in the primary outcome (TSRT and its velocity sensitivity). Between-group comparisons at individual time points (baseline, during treatment, post-intervention, and 4-week follow-up) will be performed using independent-samples Student's t-tests for normally distributed continuous variables and Mann–Whitney U tests for non-normally distributed data. Categorical variables will be compared using the Chi-square test or Fisher's exact test, as appropriate.

To evaluate treatment effects over time, repeated-measures analyses will be conducted. For normally distributed continuous outcomes, repeated-measures ANOVA will be performed to examine whether changes over time differ between treatment groups. If the assumption of sphericity is violated, Greenhouse–Geisser corrections will be applied. For non-normally distributed outcomes, appropriate non-parametric repeated-measures approaches will be used.

In addition, multivariable linear regression models will be used to explore associations between treatment allocation and changes in the primary outcome (TSRT), adjusting for clinically relevant covariates such as age, sex, time since stroke, and baseline spasticity severity, which will be predefined a priori. Study site (country) will also be included as a covariate to account for the multicenter design. Time since stroke may also be explored as a stratification variable in secondary analyses (e.g., early vs. later post-stroke phases within the 0–12 month window). If required by the data structure (e.g., missing observations or imbalance across time points), linear mixed-effects models may be considered to account for within-subject correlation and covariate adjustment.

Effect sizes (Cohen's d or partial eta squared, as appropriate) and 95% confidence intervals will be reported to support interpretation of exploratory findings and to inform the design of future definitive trials. A two-sided significance level of 5% will be adopted for the primary outcome analyses. Secondary outcomes will be interpreted in an exploratory manner.

All statistical analyses will be performed using R software (R Foundation for Statistical Computing, Vienna, Austria).

### Registration and ethics

The study is registered in Clinical Trials (NCT06296082) and Clinical Trials Information System (CTIS) under the number 2024-510866-18-00. Ethical approval was obtained in Spain, Canada, and Belgium. In Spain and Belgium, approval was granted by the Commissie Medische Ethiek GZA Ziekenhuizen (Oosterveldlaan 22, 2610 Wilrijk, Belgium; EU CT number: 2024-510866-18-00).

In Canada, ethical approval was obtained from the Research Ethics Committee of Rehabilitation and Physical Disability (RDP) of the CCSMTL, Integrated University Health and Social Services Centre for South-Central Montreal (approval number: MP-50-2023-1784; date: 30 May 2023).

All participants will provide written informed consent prior to enrolment. For individuals unable to provide consent due to cognitive or communication impairments, written informed consent will be obtained from a legally authorized representative in accordance with local regulations. No waiver of informed consent will be granted.

Any substantial modifications to the trial protocol will be submitted for prior approval to the Research Ethics Committees of all participating countries and to the relevant regulatory authorities.

Recruitment for the STROKE-POC trial began in Canada on September 10, 2024, in Belgium on September 23, 2024, and in Spain on May 2, 2025. Recruitment is currently ongoing in all three participating countries. We expect participant recruitment to be completed by June 2026, with data collection finalized by December 2026. Preliminary analyses will be conducted in early 2027, and the main results are expected to be disseminated by mid-2027.

All updates to the study protocol or timeline will be promptly reflected in the trial registries (ClinicalTrials.gov and CTIS) and communicated to all investigators, sponsors, and the Data Monitoring Committee before implementation.

## Consent for participation and ancillary studies

Written informed consent will be obtained from all participants by trained clinical investigators at each recruiting site prior to enrollment. The process will include providing an information sheet in clear language, allowing sufficient time for questions, and obtaining signatures on two copies of the consent form (one for the participant, one for the trial file). For individuals unable to provide consent due to cognitive or communication impairments, consent will be sought from an authorized legal representative in accordance with local regulations. No additional biological specimens will be collected for ancillary studies.

The informed consent obtained from participants will cover the use of de-identified clinical and outcome data exclusively for the purposes described in this protocol. Any future ancillary studies requiring participant data or biological specimens will undergo separate ethics approval and will require additional participant consent.

## Data monitoring body

An independent committee will review the study for accuracy and protocol adherence. A scientific expert committee will guide and supervise all study phases and a patient advisory group will seek feedback during the trial.

## Data deposition and curation

The STROKE-POC project data are publicly available on Zenodo at https://doi.org/10.5281/zenodo.20034064 (CC BY 4.0). Mean/median values for missing data will be imputed. Double data-entry and automated range checks will be implemented; details are described in the trial's Data Management Plan available on request.

## Safety considerations

In addition to the assessment of patient preferences and treatment acceptance, the frequency and severity of adverse effects related to DN or BTX-A will be systematically recorded.

This study is classified as a "low-intervention clinical trial" because BTX-A is authorized for use, and DN has been proven to be safe. Should one treatment be found less beneficial during the study, participants might experience reduced effects compared to the other treatment; however, this difference will be limited to the study period, with patients resuming their usual treatment afterward.

Serious adverse effects will be reviewed and managed as needed, with any severe and unexpected reactions reported promptly to health authorities and ethics committees. An annual safety report, detailing all relevant information, will be prepared by the principal investigator and promoter and submitted to these committees and authorities.

## Publications

The project's research is expected to result in several scientific publications, including studies on the comparative mechanisms and impacts of DN and BTX A treatments on spasticity, functional mobility, and quality of life, as well as their effects on gait assessment, cost-effectiveness, and muscle architecture.

## Discussion

Both BTX-A [6] and DN [7] have shown effectiveness in improving post-stroke spasticity. However, the use of DN in clinical practice is still limited despite its fewer side effects [40] and lower cost [41]. A feasibility trial directly comparing these interventions may enhance treatment prescription and provide essential data to inform sample sizes and design for future definitive trials. While clinical improvement remains central, it is also important to consider imaging-detectable muscular effects and the broader health-economic implications of these interventions. To date, no muscle imaging alterations comparable to those described after BTX-A injection have been reported following DN interventions in post-stroke populations

[7,40]. However, the optimal cumulative exposure necessary to maximize benefits while ensuring long-term safety remains to be established. Understanding these imaging findings alongside cost-effectiveness considerations may inform future decisions regarding session frequency, treatment duration, and resource allocation in larger definitive trials.

Although BTX-A exerts its primary effect through chemical denervation at the neuromuscular junction, both BTX-A injections and DN involve mechanical interaction with muscle tissue. Immediate changes in muscle tone observed after injection procedures may partially reflect mechanically mediated modulation. In contrast, DN involves repeated insertions using solid, non-beveled needles, potentially generating greater cumulative mechanical stimulation over time. The longitudinal design of this study may help explore differential temporal patterns between the immediate mechanical stimulation common to both procedures, the repeated mechanically mediated modulation induced by weekly dry needling sessions, and the delayed neurochemical effects of BTX-A. These patterns may manifest differently across muscular and functional outcomes, as well as central neurophysiological effects of changes in the muscle physiology that will be observed through examination of reflex excitability at the spinal level.

The Modified Ashworth Scale (MAS) does not fully capture the neurophysiological mechanisms underlying increased tone, including distinctions between spasticity and spastic dystonia, which may limit its value for individualized treatment decisions in heterogeneous post-stroke populations. The Tonic Stretch Reflex Threshold (TSRT), by directly quantifying stretch reflex excitability, offers a more physiologically grounded assessment. This may be particularly relevant in a 0–12 month post-stroke population characterized by variable recovery trajectories and mixed motor presentations. Emerging evidence, including case series reporting differential EMG-based responses to dry needling in patients with spasticity and spastic dystonia [42], further supports the need for more objective physiological measurements to better interpret treatment effects and inform future stratification strategies.

Further evidence is needed to understand the effects of different treatments on muscle architecture and echotexture in post-stroke patients [43] and to clarify their potential architectural and echotextural adaptations. Although major degenerative changes after BTX-A have primarily been described following longer follow-up periods or repeated injections, muscle adaptations detectable by ultrasound have also been reported after single administrations. In post-stroke populations, ultrasound analyses have shown reductions in gastrocnemius thickness and pennation angle following BTX-A injection [13]. Similarly, imaging and histological studies conducted in other populations have reported persistent muscular alterations months after injection [11,12]. Incorporating ultrasound assessment in the present study allows objective characterization of early muscular responses within a controlled comparative framework. Unlike categorical clinical scales like Heckmatt, which classify muscle involvement into broad severity levels, quantitative ultrasound enables granular assessment of tissue properties [44,45]. These exploratory data will help determine the optimal timing and relevance of muscle imaging outcomes in future trials with longer follow-up periods.

Poor balance, gait impairments [46], and leg strength [47] affect standing and walking in stroke survivors. Faster times in TUG and 10MWT do not necessarily reflect improvements in quality parameters like gait symmetry. Therefore, in addition to clinical scales, instrumented gait analysis is necessary to understand the underlying mechanisms of the observed outcomes [48].

People with poorer functional status and greater stroke severity have lower quality of life [49]. There is limited research regarding the cost-effectiveness of each treatment for stroke survivors [50]. Further research, including a cost-effectiveness analysis considering quality of life, is needed to address this gap and provide preliminary insights to guide future definitive studies and improve stroke survivor care.

As a proof-of-concept feasibility study, the expected findings will be limited to the pilot sample recruited across the three participating countries and should therefore be interpreted cautiously. The primary aim is not to establish definitive comparative effectiveness, but to generate preliminary estimates of variability, feasibility parameters, and effect sizes to inform the design of a future adequately powered trial. Although participant and operator blinding is not feasible, the use of blinded assessors and predominantly objective, instrument-based outcome measures reduces the risk of bias.

The multinational structure of this study represents an important strength, as it enhances external validity and provides insight into cross-country feasibility. Future definitive randomized controlled trials with larger sample sizes and broader geographic representation will be required to formally evaluate the effectiveness of BTX-A and DN. At that stage, advanced statistical approaches, such as multilevel modeling with country-level random effects, may be appropriate to account for clustering and contextual variability across healthcare systems.

## Conclusion

This trial will explore the effects of BTX-A and DN on post-stroke spasticity, resistance to passive stretching, gait, functional mobility, and muscle morphometric and densitometric variables. Safety, feasibility, quality of life, and cost-effectiveness will also be assessed. As a proof-of-concept feasibility study, the findings will provide preliminary insights to enhance the understanding of treatment approaches for post-stroke spasticity and to inform the design of future definitive trials.

## Supporting information

**S1 File. SPIRIT checklist and World Health Organization trial registration data set (WHO TRDS).**
(PDF)

**S2 File. SPIRIT and WHO trial registration data set.**
(PDF)

**S3 File. STROKE-POC without logo (1).**
(PDF)

**S4 File. DATA MANAGEMENT PLAN_ STROKE_POC.**
(PDF)

## Acknowledgments

We would like to express our sincere gratitude to all individuals and organizations that supported the development of this research protocol.

Furthermore, we acknowledge all colleagues and collaborators who provided insights and feedback in shaping this research protocol. Their expertise and input were highly valuable and have significantly enhanced the quality and rigor of our study design.

We are grateful to everyone who contributed to the development of this research protocol and look forward to continued support and collaboration in the future.

## Author contributions

**Conceptualization:** Clara Pujol-Fuentes, Samuel Fernández Carnero, Daniel Fernández Sanchís, Theodore Wein, Pablo Herrero, Wim Saeys, Mindy F. Levin.

**Data curation:** Bart Eeckhaut, Mindy F. Levin.

**Formal analysis:** Samuel Fernández Carnero, Daniel Fernández Sanchís, Pablo Herrero, Wim Saeys, Mindy F. Levin.

**Funding acquisition:** Clara Pujol-Fuentes, Pablo Herrero, Mindy F. Levin.

**Investigation:** Clara Pujol-Fuentes, Theodore Wein, Pablo Herrero, Mindy F. Levin.

**Methodology:** Clara Pujol-Fuentes, Theodore Wein, Pablo Herrero, Mindy F. Levin.

**Project administration:** Clara Pujol-Fuentes, Bart Eeckhaut.

**Resources:** Pablo Herrero.

**Software:** Bart Eeckhaut.

**Supervision:** Pablo Herrero, Mindy F. Levin.

**Validation:** Pablo Herrero.

**Writing – original draft:** Clara Pujol-Fuentes.

**Writing – review & editing:** Pablo Herrero, Mindy F. Levin.

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
