## [Decision Letter · Decision Letter 0]

30 Dec 2025

PONE-D-25-48383Spinal mechanisms and feasibility of Dry Needling versus Botulinum Toxin Type A in post-stroke lower limb spasticity: a proof-of-concept randomized clinical trial protocol (STROKE-POC).PLOS One

Dear Dr. Herrero,

Thank you for submitting your manuscript to PLOS ONE. After careful consideration, we feel that it has merit but does not fully meet PLOS ONE’s publication criteria as it currently stands. Therefore, we invite you to submit a revised version of the manuscript that addresses the points raised during the review process.

We look forward to receiving your revised manuscript.

Kind regards,

Esedullah Akaras

Academic Editor

PLOS One

Journal Requirements:

Reviewers' comments:

Reviewer's Responses to Questions

**Comments to the Author**

1. Does the manuscript provide a valid rationale for the proposed study, with clearly identified and justified research questions?

Reviewer #1: Yes

Reviewer #2: Yes

Reviewer #3: Partly

Reviewer #4: Yes

2. Is the protocol technically sound and planned in a manner that will lead to a meaningful outcome and allow testing the stated hypotheses?

Reviewer #1: Yes

Reviewer #2: Yes

Reviewer #3: Yes

Reviewer #4: Yes

3. Is the methodology feasible and described in sufficient detail to allow the work to be replicable?

Reviewer #1: Yes

Reviewer #2: Yes

Reviewer #3: Yes

Reviewer #4: Yes

4. Have the authors described where all data underlying the findings will be made available when the study is complete?

Reviewer #1: Yes

Reviewer #2: Yes

Reviewer #3: Yes

Reviewer #4: Yes

5. Is the manuscript presented in an intelligible fashion and written in standard English?

Reviewer #1: Yes

Reviewer #2: Yes

Reviewer #3: Yes

Reviewer #4: Yes

6. Review Comments to the Author

You may also provide optional suggestions and comments to authors that they might find helpful in planning their study.

Reviewer #1: The authors present a protocol for an interesting study of BTX-A in post-stroke lower limb spasticity. To be a contribution, a pre-enrollment protocol must be sufficiently rich in background information, precise in inclusion criteria, and thorough enough in presenting the rationale for choosing outcome measures. I suggest that the authors explicitly use the SPIRIT 2025 guidelines in preparing a revised manuscript, if the Editor requests one.

Now for a disclaimer: Prior to my retirement, I had considerable experience in using BTX-A for treatment of spasticity and sometimes pain in children and adolescents with cerebral palsy (CP ) and, less frequently, with stroke. I had no intentional experience with dry needling. However, I did notice an immediate reduction in tone immediately after injections many times. In retrospect, this was probably a consequence of dry needling, although it was not recognized as such at the time by me or by others.

Considering study background first, a thorough review of the literature about dry needlingly would be helpful for the uninformed reader, like I was. Some specific questions that the authors might consider addressing include: What is the most likely mechanism of action of dry needling for tone reduction? How did you arrive at the weekly frequency of dry needling? Was it based on your personal experience? In your experience or the literature, what is the longest dry needling can be continued without muscle damage? A more thorough summary of the literature on BTX in strokes would also be helpful.

Considering inclusion criteria next, please comment in the manuscript on why you decided not to limit the study population to hemiplegic stroke subjects. This would have made the study population more homogeneous. Many RTCs of a treatment in stroke have taken this approach. At the least, the greater homogeneity of the study population from limiting subjects to those with hemiplegia would enable comparison of your findings to those of other interventions in same population.

It is possible that the duration of the interval between the stroke and treatment may affect outcomes. In my experience, patients with stroke sometimes developed dystonia as a late complication (although this may be more common in childhood than in your subjects). For subjects enrolled later in the enrollment period (which, for the Editor, is between the stroke and 12 months after the stroke), do you think that the development of dystonia accompanying spasticity may confound your functional outcomes? It certainly did in my patients with both spasticity and dystonia from stroke. How do you plan to handle this potential confounding variable? Separately, for subjects enrolled very early in the enrollment period, do you think that evolution of tone and/or neurological recovery independent of treatment will affect outcomes and if not, why not? Do you have a plan to identify and address this potential confounder? Although each subject will serve as his/her own control, it would be helpful to know about the effects on outcomes of both of these potential confounders to guide future studies.

Considering outcome variables, do you think that your quality-of-life measure is going to be more useful for this lower extremity intervention than a more specific ambulation outcome, such steps per day in activities of daily living, assessed using an accelerometer? I think not. Please refute this suggestion if you decide to stick with a QoL outcome. Please also present studies that validate the outcomes measures obtained using the Montreal Spasticity Reflex Threshold System.

Do you expect to see much clinically significant change from BTX-A on muscle ultrasound at 1month? In CP at least, it takes time to damage the muscle from BTX-A injections. As one BTX-A skeptic memorably said at an international meeting, “Botox injections will eventually swap a muscle condition for a muscle disease”.

You state that your multivariate regression models will” include significant variables”. What specific covariates do you plan to measure and then adjust for in your regression analyses? Finally, you will likely have done too many outcome analyses for any marginally significant difference to withstand Bonferroni correction. Thoughts about this?

Reviewer #2: 1. BTX-A was administered as a single injection, while DN was given once weekly for 12 weeks. Could this design affect the comparability of the results? 2. While assessor blinding is feasible, have the potential effects of the lack of blinding for both patients and operators been considered?

Reviewer #3: This manuscript is essentially a study protocol to conduct a proof-of-concept (PoC), feasibility, multi-country, randomized clinical trial (RCT) to asses the differences between the treatments: (a) one session of BTXA (Botulinum Toxin type A), and (b) 12 weekly sessions of DN (Direct Needling), in reducing post-stroke lower limb spasticity. The study was registered within the ClinicalTrials.gov, and approved by the respective Ethics/IRB board. While the objectives and timeliness of this project appear sound and convincing, some comments appear below, following CONSORT guidelines and statistical perspectives:

(a) Details on randomization are needed. How are allocation concealment and blinding done? They are not the same thing; the distinction needs to be articulated, keeping in the context.

(b) PoC trials are mostly exploratory; hence a formal sample size/power calculation can be excused. Having said that, some kind of formal sample size justification, using targeted effect sizes (say, moderate), is still necessary (see here: https://pmc.ncbi.nlm.nih.gov/articles/PMC4876429/), which the manuscript is lacking. The corresponding sample size/power statements should mention the name of the statistical test, level of significance, and other necessary parameters. It should also be justified using the primary endpoint.

(c) Statistical/Quantitative Analysis:

(c1) Analysis needs to be conducted separately (at each time point), and then combining all points, to assess the differences between the two treatments. The writeup should mention which test will be appropriate for which situation.

(c2) It's not clear, what the authors meant by ANOVA test. There are 2 groups, where a Student-t, or a Wilcoxon rank-sum will suffice. ANOVA will be helpful for testing > 2 groups, which is likely not the case as per my understanding, at a specific time point. For testing 2 groups (combining multiple time points), a repeated measures ANOVA (or, it's nonparametric counterpart) will be helpful.

(c3) During the trial, repeated measures ANOVA (or a Friedman's test) can be employed. However, additional subject-level covariates will also be collected. Why was a formal linear mixed model, or a GEE not planned, since the total proposed sample size of 90 doesn't look so bad. The ANOVA precludes assessment of group differences without factoring in the covariates, which the mixed models, or GEE can factor in easily.

(c4) It's appealing to mention the name of the software to be used to conduct the analysis.

(c5) The sentence "Multivariate regression models will include significant variables..." appears vague. First, the word multivariate is used, when one analyzes multiple outcomes at the same time, adjusted for multiple predictors. Are the authors doing that? Guess not. The correct word can be "multivariable" regression, which implies one response/outcome, regressed upon multiple predictors.

Furthermore, it's not clear what the authors meant by significant variables? Will they be conducting a variable selection? It's not clear. Even before that, how many variables/covariates the trial may generate, prompting them to conduct variable selection? More details needed.

(d) Conclusions/Discussion: This section should mention that the expected results from this study are only valid for this pilot sample of patients recruited, and should allude to future trials with much larger sample sizes varying with geographical locations to assess the effectiveness of the competing treatments. The authors need to be commended for at least conducing a 3 country RCT, the scope of which can be extended to multiple countries, depending on resources. At that future stage, innovative analysis are required, alluding to multi-level modeling (say, putting country-level random effects). This can also be part of the Conclusion/Discussion sections.

Reviewer #4: This protocol paper presents a multicenter, randomized clinical trial comparing Dry Needling (DN) and Botulinum Toxin Type A (BTX-A) for post-stroke lower limb spasticity. The study aims to elucidate spinal mechanisms, feasibility, safety, and exploratory effects at muscular, functional, quality-of-life, and cost-effectiveness levels. Ninety participants from Canada, Belgium, and Spain will be randomized to receive either a single BTX-A session or twelve weekly DN sessions, with outcomes assessed by blinded evaluators.

Strengths of the study

The trial addresses a significant gap: no previous studies have directly compared DN and BTX-A across spinal, muscular, functional, and economic domains in post-stroke spasticity. This is highly relevant given the prevalence of spasticity and the limitations of current treatments. The inclusion of cost-effectiveness and quality-of-life outcomes is forward-thinking, reflecting real-world concerns for patients and healthcare systems. The protocol follows SPIRIT and WHO guidelines, with ethical approval in all participating countries and registration in major trial registries. Blinded outcome assessment and stratified randomization enhance internal validity. The use of objective measures (e.g., Tonic Stretch Reflex Threshold, instrumented gait analysis, ultrasound imaging) is a methodological strength.

Weaknesses of the study

The comparison between a single BTX-A session and twelve DN sessions may introduce bias due to unequal treatment exposure. While this reflects clinical practice, it complicates direct efficacy comparisons and may affect cost-effectiveness analyses.

The sample size (90 participants) is justified for feasibility but is not powered for efficacy. This limits the ability to draw definitive conclusions about clinical effectiveness, especially for secondary outcomes. Outcome assessors are blinded, participants and clinicians are not, due to the nature of the interventions. This could introduce performance and detection bias, particularly for subjective outcomes like quality of life. Inclusion criteria (e.g., first stroke, MAS 1-2, independent ambulation) may exclude more severe cases, limiting applicability to the broader stroke population. The protocol references a Standard Operating Procedure (SOP) and Data Management Plan, but these documents are not included. Full transparency would benefit from their publication as supplementary materials.

Future Studies

The study’s findings will provide valuable preliminary data for larger, definitive trials. Future research should consider: Powering for efficacy and including broader patient populations; Exploring long-term outcomes and repeated treatment cycles; Investigating patient-centered outcomes and preferences in greater depth; Publishing detailed SOPs and data management plans for reproducibility.

Conclusion

This protocol represents a well-designed, clinically relevant feasibility trial that addresses a critical gap in post-stroke spasticity management. Its strengths lie in methodological rigor, innovative outcome measures, and multinational collaboration. However, limitations in intervention comparability, sample size, and blinding should be considered when interpreting future results.

7. PLOS authors have the option to publish the peer review history of their article (what does this mean?). If published, this will include your full peer review and any attached files.

Reviewer #1: No

Reviewer #2: No

Reviewer #3: No

Reviewer #4: No

---

## [Author Response · Author response to Decision Letter 1]

1 Apr 2026

The responses can be found in the "Response to reviewers" document.

---

## [Decision Letter · Decision Letter 1]

19 Apr 2026

Spinal mechanisms and feasibility of Dry Needling versus Botulinum Toxin Type A in post-stroke lower limb spasticity: a proof-of-concept randomized clinical trial protocol (STROKE-POC).

PONE-D-25-48383R1

Dear Dr. Herrero,

We’re pleased to inform you that your manuscript has been judged scientifically suitable for publication and will be formally accepted for publication once it meets all outstanding technical requirements.

Kind regards,

Esedullah Akaras

Academic Editor

PLOS One

Additional Editor Comments (optional):

Reviewers' comments:

Reviewer's Responses to Questions

**Comments to the Author**

1. Does the manuscript provide a valid rationale for the proposed study, with clearly identified and justified research questions?

Reviewer #3: Yes

2. Is the protocol technically sound and planned in a manner that will lead to a meaningful outcome and allow testing the stated hypotheses?

Reviewer #3: Yes

3. Is the methodology feasible and described in sufficient detail to allow the work to be replicable?

Reviewer #3: Yes

4. Have the authors described where all data underlying the findings will be made available when the study is complete?

Reviewer #3: Yes

5. Is the manuscript presented in an intelligible fashion and written in standard English?

Reviewer #3: Yes

6. Review Comments to the Author

You may also provide optional suggestions and comments to authors that they might find helpful in planning their study.

Reviewer #3: The authors were able to address my previosu round of comments with satisfaction. I have no additional comments.

7. PLOS authors have the option to publish the peer review history of their article (what does this mean?). If published, this will include your full peer review and any attached files.

Reviewer #3: No

---

## [Editor Report · Acceptance letter]

PONE-D-25-48383R1

PLOS One

Dear Dr. Herrero,

I'm pleased to inform you that your manuscript has been deemed suitable for publication in PLOS One. Congratulations! Your manuscript is now being handed over to our production team.

Kind regards,

on behalf of

Dr. Esedullah Akaras

Academic Editor

PLOS One